# Altercrasins A–E, Decalin Derivatives, from a Sea-Urchin-Derived *Alternaria* sp.: Isolation and Structural Analysis Including Stereochemistry

**DOI:** 10.3390/md17040218

**Published:** 2019-04-11

**Authors:** Takeshi Yamada, Asumi Tanaka, Tatsuo Nehira, Takumi Nishii, Takashi Kikuchi

**Affiliations:** 1Department of Medicinal Molecular Chemistry, Osaka University of Pharmaceutical Sciences, 4-20-1, Nasahara, Takatsuki, Osaka 569-1094, Japan; ichigo-ame.xxx@ezweb.ne.jp (A.T.); n.t.rokusyo@i.softbank.jp (T.N.); t.kikuchi@gly.oups.ac.jp (T.K.); 2Graduate School of Integrated Arts and Sciences, Hiroshima University, 1-7-1 Kagamiyama, Higashi-Hiroshima 739-8521, Japan; tnehira@hiroshima-u.ac.jp

**Keywords:** altercrasins, *Alternaria* sp., *Anthocidaris crassispina*, decalin derivatives, cytotoxicity

## Abstract

In order to find out the seeds of antitumor agents, we focused on potential bioactive materials from marine-derived microorganisms. Marine products include a number of compounds with unique structures, some of which may exhibit unusual bioactivities. As a part of this study, we studied metabolites of a strain of *Alternaria* sp. OUPS-117D-1 originally derived from the sea urchin *Anthocidaris crassispina*, and isolated five new decalin derivatives, altercrasins A–E (**1**–**5**). The absolute stereostructure of altercrasins A (**1**) had been decided by chemical transformation and the modified Mosher’s method. In this study, four decalin derivatives, altercrasins B–E (**2**–**5**) were purified by silica gel chromatography, and reversed phase high-performance liquid chromatography (RP HPLC), and their structures were elucidated on the basis of 1D and 2D nuclear magnetic resonance (NMR) spectroscopic analyses. The absolute configuration of them were deduced by the comparison with **1** in the NMR chemical shifts, NOESY correlations, and electronic circular dichroism (ECD) spectral analyses. As a result, we found out that compound pairs of **1**/**2** and **4**/**5** were respective stereoisomers. In addition, their cytotoxic activities using murine P388 leukemia, human HL-60 leukemia, and murine L1210 leukemia cell lines showed that **4** and **5** exhibit potent cytotoxicity, in especially, the activity of **4** was equal to that of 5-fluorouracil.

## 1. Introduction

Marine organisms are a potential prolific source of highly bioactive secondary metabolites with unique structures that may serve as useful seeds for the development of new chemotherapy agents [1,2]. Previously, our group has focused on potential new antitumor materials from marine-derived microorganisms that produce several compounds bearing unique structures [3,4,5]. As a part of this study, metabolites from the fungus *Alternaria* sp. OUPS-117D-1 originally obtained from the sea urchin *Anthocidaris crassispina* were examined, and a new compound designated as altercrasin A (**1**) (Figure 1) was isolated. As has been reported previously, **1** was a cytochalasin-like decalin derivative with spirotetramic acid [6]. Previous reports have also introduced delaminomycins [7] isolated from *Streptomyces albulus*; lucensimycins [8,9,10] isolated from *Streptomyces lusensis*, fusarisetin A [11,12,13] isolated from *Fusarium* sp., and diaporthichalasin [14,15] isolated from *Diaporte* sp. as metabolites with a similar decalin derivative. The absolute configuration of this class with a spiro-lactam or -lactone cannot be experimentally determined unless a good single crystal is obtained [11]. However, in a previously reported study for the absolute stereostructure of **1**, our group has successfully determined it by experiments via a chemical transformation [6]. Our continuous search for cytotoxic metabolites from this fungal strain afforded four new decalin derivatives designated as altercrasins B–E (2–5), respectively (Figure 1). As these were minor components, the above-mentioned method was not used to elucidate the stereostructures. In this study, the chiral centers in these metabolites were assigned by NMR and ECD spectral analyses.

## 2. Results and Discussion

*Alternaria* sp., a microorganism from *A. crassispina*, was cultured at 27 °C for 6 weeks in a medium (70 L) containing 1% glucose, 1% malt extract, and 0.05% peptone in artificial seawater adjusted to pH 7.5. After incubation, the EtOAc extract of the culture filtrate was purified via bioassay-directed fractionation by using a stepwise combination of silica-gel column and Sephadex LH-20 chromatography, followed by reverse-phase HPLC, affording altercrasins A (**1**) (10.3 mg, 0.020%), B (**2**) (6.5 mg, 0.012%), C (**3**) (4.6 mg, 0.009%), D (**4**) (1.3 mg, 0.002%), and E (**5**) (5.3 mg, 0.010%) as pale yellow oils.

As it has been observed for previously reported **1** [6], altercrasin B (**2**) also exhibited the molecular formula C_24_H_33_NO_5_ as established by the [M + Na]^+^ peak in HRFABMS. The IR spectrum exhibited absorption bands at 3478, 1720, and 1710 cm^−1^ characteristic of hydroxyl and carbonyl groups. The close inspection of the ^1^H and ^13^C NMR spectra of **2** (Table 1 and Appendix A) by using distortionless enhancement by polarization transfer (DEPT) and ^1^H–^13^C correlation spectroscopy (HMQC) revealed the presence of two secondary methyl groups (C-17 and C-23, respectively); a tertiary methyl (C-24) group; three sp^3^-hybridized methylene groups (C-1, C-2, and C-4, respectively); six sp^3^-methine groups (C-3, C-5, C-8, C-10, C-13, and C-16, respectively); two quaternary sp^3^-carbon groups (C-9 and C-12, respectively); four sp^2^-methine groups (C-6, C-7, C-14, and C-15, respectively); and three carbonyl groups (C-11, C-18, and C-22, respectively), including an amide carbonyl (C-22). The ^1^H–^1^H COSY analysis of **2** led to three partial structures, including a hydroxy butylene group (H-13/H-14, H-14/H-15, H-15/H-16, H-16/16-OH, and H-16/H-17) and a hydroxyethyl group (H-19/H-20, H-20/20-OH, and H-20/H-21), as indicated by the bold-faced lines shown in Figure 2. The key HMBC correlations shown in Figure 2 verified the connection of these three units and the remaining functional groups, indicating that the planar structure of **2** is the same as that of **1**.

The stereochemistry of **2** was deduced from NOESY experiments and the comparison of NMR and ECD spectral data with **1**, where the absolute configuration of **1** was determined by the application of the modified Mosher’s method after the NOESY experiment of the acetonide derivative following the stereoselective reduction at the carbonyl C-18 group in **1**.^4^ The observed NOESY correlations of **2** (H-1β/H-3, H-1β/H-5, H-1β/H-24, H-5/H-24, H-8/H-14, H-8/H-24, and H-10/H-13) revealed that the relative configuration of the decalin moiety is the same as that of **1**; i.e., **2** was a diastereomer of **1** at C-12, C-16, C-19, or C-20. In ^1^H NMR spectrum of **2** (Table 1 and Appendix A), the general features closely resembled those of **1**, except for the ^1^H NMR signals for H-19 (**1**; *δ*_H_ 3.66 and **2**; *δ*_H_ 3.93)). In addition, the coupling constant for the H-19 signal was different (**1**; *J*_19,20_ = 6.8 Hz and **2**; *J*_19,20_ = 4.2 Hz). These differences were supposedly related to the magnetic anisotropic effect of the carbonyl group (C-18) that reflected the change of a dihedral angle between H-19 and H-20, indicating that the absolute configuration at C-19 in **2** is opposite to that of **1**. The chemical shifts for the ^1^H and ^13^C NMR signals at C-16 and C-20 in **2** resembled those of **1**, indicative of identical chirality at C-16 and C-20 between **1** and **2** (Table 1). On the other hand, the ECD Cotton effects were expected to be induced by the carbonyl groups (C-11, C-18, and C22) around C-12. Hence, the experimental observation that the ECD spectrum of **2** shows good agreement with that of **1** revealed identical chirality at C-12 between **1** and **2** (Figure 3). The above evidence suggested that **2** is an epimer of **1** at C-19.

According to HRFABMS data, altercrasin C (**3**) was assigned the molecular formula C_22_H_27_NO_4_. The general features of its IR spectrum matched those of **1** and **2**. By the accurate inspection of the NMR spectrum of **3**, the signals corresponding to the hydroxyethyl moiety (C-20–C-21) and the carbonyl (C-18) group observed in **1** and **2** were replaced by amide carbonyl (C-19 (*δ*_C_ 173.3)) and a quaternary sp^3^ carbon groups with a hydroxyl group (C-18 (*δ*_C_ 81.2)) (Table 1 and Appendix A). In addition, the geometrical configuration of the side-chain olefin moiety (C-14 to C-15) as *cis* from the ^1^H NMR coupling constant (*J*_14, 15_ 8.8 Hz) was revealed to be less than that of **1** (*J*_14, 15_ 15.5 Hz). The above evidences and the ^1^H–^1^H COSY and HMBC correlation shown in Figure 4 and Appendix A led to the planar structure of **3**, which had a cyclic imide. For the stereochemistry of **3**, the NOESY correlations (H-1/H-3, H-1/H-5, H-1/H-24, H-4/H-10, H-5/H-24, H-8/H-24, and H-10/H-13) for the decalin moiety revealed that the relative configurations among C-3, C-5, C8, C-9, C-10, and C-13 are the same as **1** and **2**. In addition, the NOESY correlation between H-13 and H-17 revealed that the 16-methyl group is oriented *cis* to H-13 (Figure 5). The absolute configurations in the decalin moiety were hypothesized as those of the above compounds in terms of the biosynthetic pathway. To determine the absolute configuration of the remaining chiral centers (C-12 and C-18), the ECD spectrum of **3** following a conformational consideration by the NOESY experiment and the building of an HGS molecular model was recorded. For all possible combinations, **3a** (12*S*, 18*R*), **3b** (12*S*, 18*S*), **3c** (12*R*, 18*R*), and **3d** (12*R*, 18*S*) (Figure 6), the *trans* ring junction at C-12 and C-18 such as **3a** and **3d** led to a large distortion of the five-membered cyclic imide; hence, the possibility of **3** exhibiting the **3a** and **3d** stereostructures is excluded in addition to some contradiction of NOESY correlation; i.e., the correlation between H-17 and H-13 should not be observed in **3a**. In ECD spectra of **1**, **2**, and **3**, the Cotton effect at 285 nm corresponded to the carbonyl groups (*n→π** interaction); hence, the absolute configuration at C-12 is assigned (Figure 3). The ECD spectra of **3** showed a negative Cotton effect at 285 nm; hence, the absolute configuration at C-12 in **3** is deduced to be 12*R*, i.e., **3** exhibited the stereostructure of **3c**.

Altercrasins D (**4**) and E (**5**) were assigned the same molecular formula C_24_H_31_NO_4_ based on the deduction according to HRFABMS data. ^1^H and ^13^C NMR spectra of **4** and **5** revealed similar features except that the differences for the chemical shift between **4** (proton signals: H-1α (*δ*_H_ 1.67), H-10 (*δ*_H_ 1.70), H-13 (*δ*_H_ 3.71), H-14 (*δ*_H_ 5.52), and H-19 (*δ*_H_ 3.49); carbon signals: C-11 (*δ*_C_ 208.4), C-12 (*δ*_C_ 70.5), C-13 (*δ*_C_ 43.6), C-14 (*δ*_C_ 124.6), C-18 (*δ*_C_ 208.4), and C-22 (*δ*_C_ 168.1)) and **5** (proton signals: H-1α (*δ*_H_ 1.91), H-10 (*δ*_H_ 1.37), H-13 (*δ*_H_ 3.96), H-14 (*δ*_H_ 5.70), and H-19 (*δ*_H_ 3.81); carbon signals: C-11 (*δ*_C_ 205.2), C-12 (*δ*_C_ 72.7), C-13 (*δ*_C_ 41.5), C-14 (*δ*_C_ 126.9), C-18 (*δ*_C_ 204.7), and C-22 (*δ*_C_ 170.7)) were observed (Table 2, Appendix A). Analyses of HMBC correlations confirmed that **4** and **5** have the same planar structure (Figure 7). In the NOESY experiments of **4** and **5** (Table 2, Appendix A), the observed correlations clearly indicated their homologous stereochemistry except for the spiro γ-lactam moiety; i.e., diastereomeric relationship at C-12, C-19, or C-20 between **4** and **5** was revealed.

Their absolute stereostructure was considered to be based on the relationship between the ECD Cotton effects and the absolute configuration at C-12 as described above. The Cotton effects at 285 nm in the CD spectra of **4** and **5** clearly revealed the absolute configuration at C-12, i.e., **4** possessed 12*S*, and **5** possessed 12*R* (Figure 8). For the ^1^H and ^13^C NMR signals at C-20 in **4** and **5**, the chemical shifts were in good agreement with those of **1** and **2**. Based on this evidence and the fact that these were isolated from the same fungal metabolites, we guessed that **4** and **5** exhibit a 20*R* configuration as **1** and **2**. On the other hand, the ^1^H NMR signals for H-19 in **4** and **5** demonstrated the same difference in the chemical shifts and spin–spin coupling constants (**4**: *δ*_H_ 3.49 (7.2 Hz) and **5**: *δ*_H_ 3.81 (4.8 Hz)) as those of **1** and **2** (Table 1 and Table 2); hence, the dihedral angle between H-19 and H-20 in **4** and **5** is thought to be the same as those of **1** and **2**, respectively, i.e., **4** and **5** exhibited the 19*S* and 19*R* configurations, respectively. Thus, the absolute configuration of **4** (12*S*,19*S*,20*R*) and **5** (12*R*,19*R*,20*R*) is proposed. These stereostructural hypotheses will be positively supported by synthetic research or X-ray crystal structure analysis in the future.

In this study, the cancer cell growth inhibitory property of the metabolites was examined using murine P388 leukemia, human HL-60 leukemia, and murine L1210 leukemia cell lines. Table 3 summarizes the results. **4** and **5** bearing a diene moiety (C-6 to C-8) exhibited significant cytotoxic activity against these cancer cells. In particular, the activity of **4** was equal to that of 5-fluorouracil. Contrary to our expectations, the difference in the stereochemistry was not related to the activity. Currently, investigation of related compounds isolated from this fungal metabolite is underway, as well as their structure-activity relationships.

## 3. Materials and Methods

### 3.1. General Experimental Procedures

NMR spectra were recorded on an Agilent-NMR-vnmrs (Agilent Technologies, Santa Clara, CA, USA) 600 with tetramethylsilane (TMS) as an internal reference. FABMS was recorded using a JEOL JMS-7000 mass spectrometer (JEOL, Tokyo, Japan). IR spectra was recorded on a JASCOFT/IR-680 Plus (Tokyo, Japan). Optical rotations were measured using a JASCO DIP-1000 digital polarimeter (Tokyo, Japan). Silica gel 60 (230–400 mesh, Nacalai Tesque, Inc., Kyoto, Japan) was used for column chromatography with medium pressure. ODS HPLC was run on a JASCO PU-1586 (Tokyo, Japan) equipped with a differential refractometer RI-1531 (Tokyo, Japan) and Cosmosil Packed Column 5C18-MSII (25 cm × 20 mm i.d., Nacalai Tesque, Inc., Kyoto, Japan). Analytical TLC was performed on precoated Merck aluminum sheets (DC-Alufolien Kieselgel 60 F254, 0.2 mm, Merck, Darmstadt, Germany) with the solvent system CH_2_Cl_2_-MeOH (19:1) (Nacalai Tesque, Inc., Kyoto, Japan), and compounds were viewed under a UV lamp (AS ONE Co., Ltd., Osaka, Japan) and sprayed with 10% H_2_SO_4_ (Nacalai Tesque, Inc., Kyoto, Japan) followed by heating. 

### 3.2. Fungal Material

A strain of *Alternaria* sp. OUPS-117D-1 was initially isolated from the sea urchin *Anthocidaris crassispana*, collected in Osaka bay, Japan in May 2010. The fungal strain was identified by Techno Suruga Laboratory Co., Ltd., Shizuoka, Japan. The sea urchin, which wiped its surface with EtOH, was cracked by a scalpel and its inside was applied to the surface of nutrient agar layered in a Petri dish. Serial transfers of one of the resulting colonies provided a pure strain of *Alternaria* sp. 

### 3.3. Culturing and Isolation of Metabolites

The fungal strain was cultured at 27 °C for six weeks in a liquid medium (70 L) containing 1% glucose, 1% malt extract and 0.05% pepton in artificial seawater adjusted to pH 7.5. The fungal strain filtrated culture broth was extracted thrice with MeOH (Nacalai Tesque, Inc., Kyoto, Japan). The combined extracts were evaporated in vacuo to afford a mixture of crude metabolites (52.8 g). The MeOH extract was chromatographed on a silica gel (Nacalai Tesque, Inc., Kyoto, Japan) column with a CH_2_Cl_2_-MeOH (Nacalai Tesque, Inc., Kyoto, Japan) gradient as the eluent. The MeOH-CH_2_Cl_2_ (5:95) eluate (22.6 g) was chromatographed again on a silica gel column with a hexane-EtOAc-MeOH (Nacalai Tesque, Inc., Kyoto, Japan) gradient as the eluent. The MeOH–EtOAc (2:98 and 5:95) eluate (F1 (1.9 g), F2 (1.3 g), respectively) were chromatographed on LH-20 (GE Healthcare Japan, Tokyo, Japan) using MeOH-CH_2_Cl_2_ (1:1) as the eluent. The fraction exhibiting cytotoxicity from F1 (105.1 mg) was purified by HPLC using MeOH-H_2_O (80:20) as the eluent to afford **3** (4.6 mg), **4** (1.3 mg), and **5** (5.3 mg). The fraction exhibiting cytotoxicity from F2 (794.5 mg) was purified by HPLC using MeOH-H_2_O (80:20) as the eluent to afford **1** (10.3 mg) and **2** (6.5 mg).

Altercrasin A (**1**). colorless oil; IR (film) *ν*_max_ 3345, 1730 1671 cm^−1^; NMR data, see Table 1 and Appendix A, and the previous report^4^; HRFABMS [M + H]^+^
*m/z* 416.2436 (calcd for C_24_H_34_NO_5_: 416.2437).

Altercrasin B (**2**). colorless oil; IR (film) *ν*_max_ 3368, 1729 1701 cm^−1^; NMR data, see Table 1 and Appendix A; HRFABMS [M + H]^+^
*m/z* 416.2430 (calcd for C_24_H_34_NO_5_: 416.2437).

Altercrasin C (**3**). colorless oil; IR (film) *ν*_max_ 3478, 1720 1710 cm^−1^; NMR data, see Table 1 and Appendix A; HRFABMS [M + Na]^+^
*m/z* 392.1839 (calcd for C_22_H_27_NO_4_Na: 392.1837).

Altercrasin D (**4**). colorless oil; IR (film) *ν*_max_ 3318, 2917, 1693 cm^−1^; NMR data, see Table 2 and Appendix A; HRFABMS [M + H]^+^
*m/z* 398.2332 (calcd for C_24_H_32_NO_4_: 398.2332).

Altercrasin E (**5**). colorless oil; IR (film) *ν*_max_ 3318, 2918 1701 cm^−1^; NMR data, see Table 2 and Appendix A; HRFABMS [M + H]^+^
*m/z* 398.2332 (calcd for C_24_H_32_NO_4_: 398.2332).

### 3.4. Assay for Cytotoxicity

Cytotoxic activities of **1**–**5** and 5-fluorouracil were examined with the 3-(4,5-dimethyl-2-thiazolyl)-2,5-diphenyl-2H-tetrazolium bromide (MTT) method. P388, HL-60, and L1210 cells were cultured in RPMI 1640 Medium (10% fetal calf serum) at 37 °C in 5% CO_2_. The test materials were dissolved in dimethyl sulfoxide (DMSO) to give a concentration of 10 mM, and the solution was diluted with the Essential Medium to yield concentrations of 200, 20, and 2 μM, respectively. Each solution was combined with each cell suspension (1 × 10^−5^ cells/mL) in the medium, respectively. After incubating at 37 °C for 72 h in 5% CO_2_, grown cells were labeled with 5 mg/mL MTT in phosphate-buffered saline (PBS), and the absorbance of formazan dissolved in 20% sodium dodecyl sulfate (SDS) in 0.1 N HCl was measured at 540 nm with a microplate reader (MTP-310, CORONA electric). Each absorbance values were expressed as percentage relative to that of the control cell suspension that was prepared without the test substance using the same procedure as that described above. All assays were performed three times, semilogarithmic plots were constructed from the averaged data, and the effective dose of the substance required to inhibit cell growth by 50% (IC_50_) was determined.

## 4. Conclusions

In this study, four new decalin derivatives designated as altercrasins B–E (**2**–**5**) were isolated from a strain of *Alternaria* sp. OUPS-117D-1 originally derived from the sea urchin *A. crassispina*. Their chemical structures were confirmed by NMR spectral analysis, and their plausible stereochemistry was deduced by considering the NMR chemical shifts and spin–spin coupling constants, as well as the assignment of ECD Cotton effects. As a result of the assay for cytotoxicity, **4** and **5** bearing a diene moiety (C-6 to C-8) exhibited significant cytotoxic activity against these cancer cells, especially the HL-60 cell line. In especially, the activity of **4** was equal to that of 5-fluorouracil.

## Figures and Tables

**Figure 1 marinedrugs-17-00218-f001:**
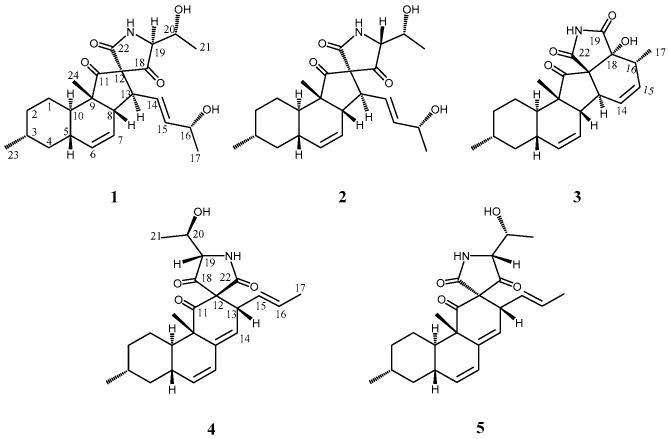
Structures of altercrasins A–E (**1**–**5**).

**Figure 2 marinedrugs-17-00218-f002:**
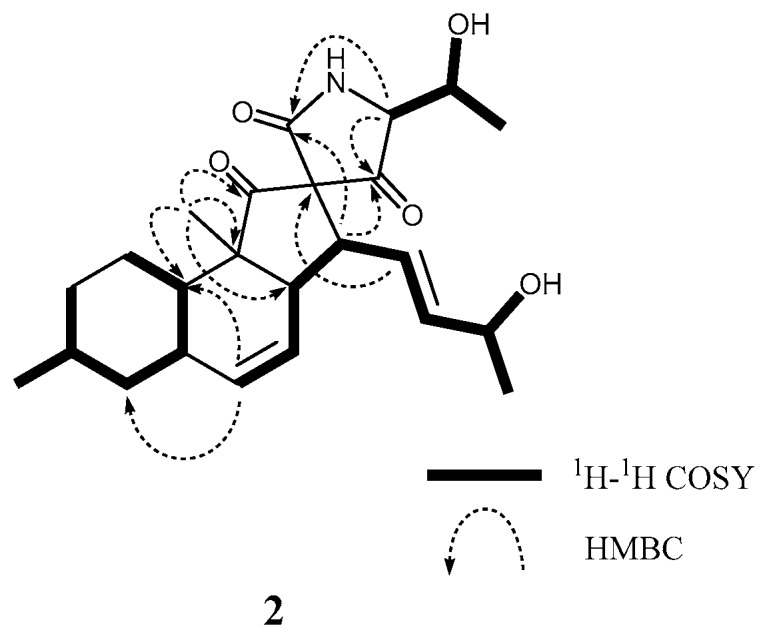
Typical 2D NMR correlations in **2**.

**Figure 3 marinedrugs-17-00218-f003:**
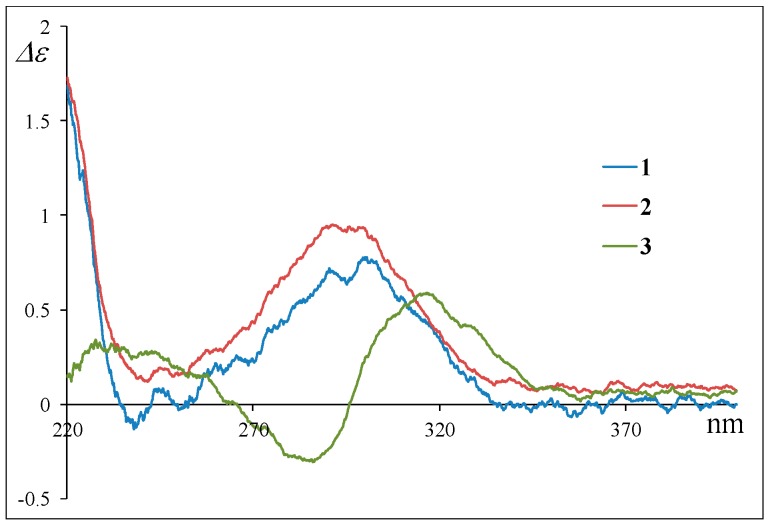
Experimental ECD spectra of **1**, **2**, and **3**.

**Figure 4 marinedrugs-17-00218-f004:**
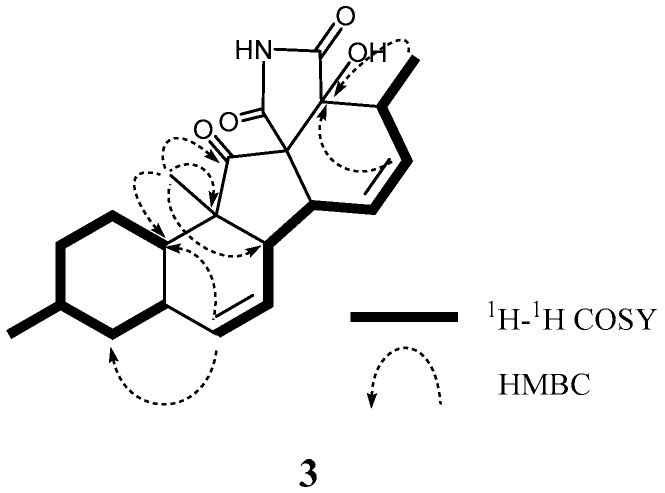
Typical 2D NMR correlations in **3**.

**Figure 5 marinedrugs-17-00218-f005:**
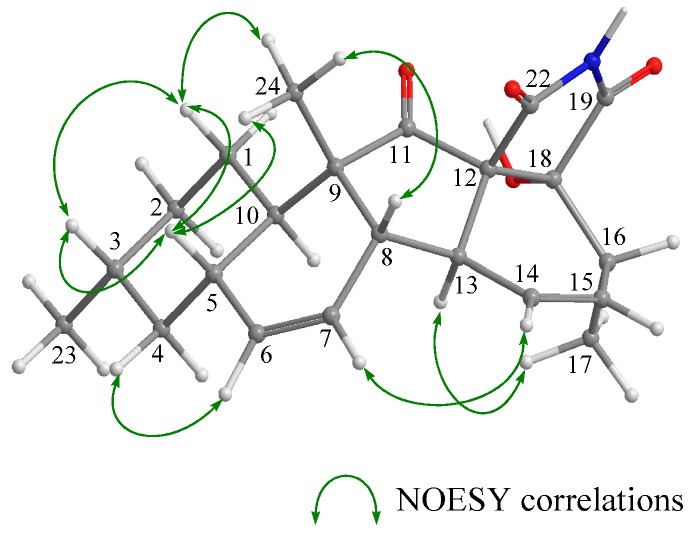
Key NOESY correlations in **3**.

**Figure 6 marinedrugs-17-00218-f006:**
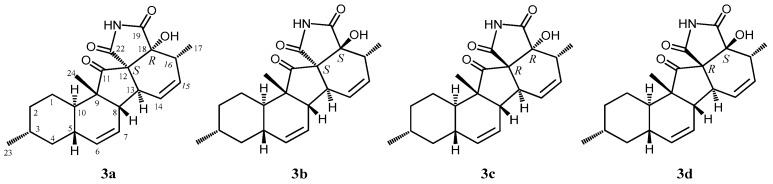
Four plausible structures of **3**.

**Figure 7 marinedrugs-17-00218-f007:**
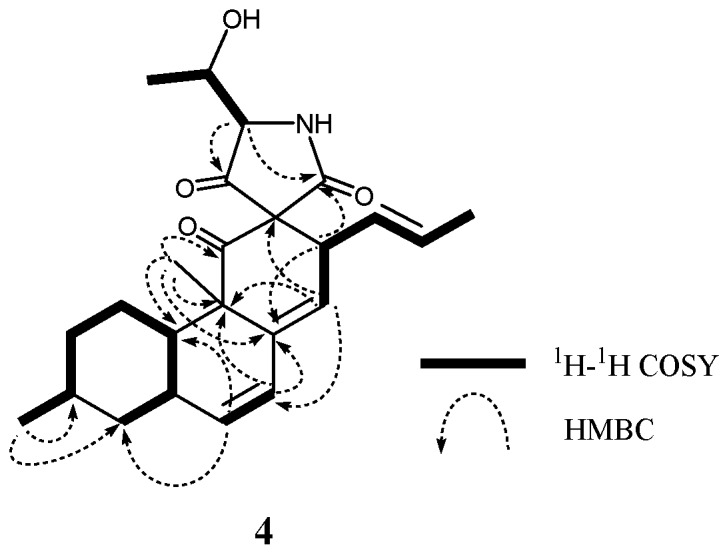
Typical 2D NMR correlations in **4**.

**Figure 8 marinedrugs-17-00218-f008:**
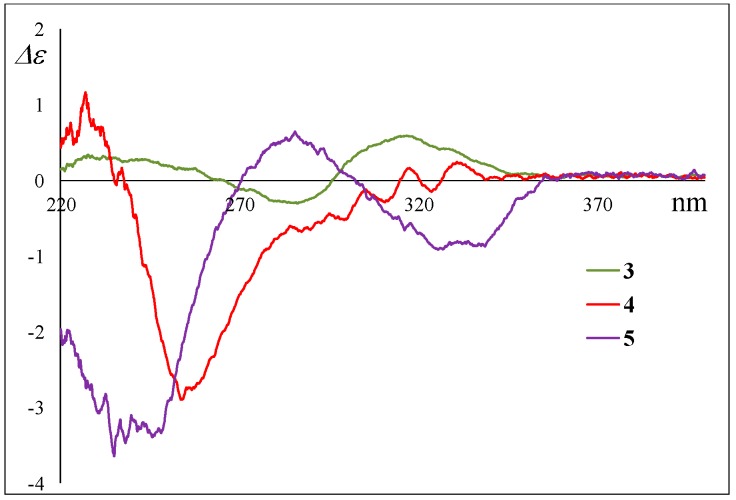
Experimental ECD spectra of **4** and **5**.

**Table 1 marinedrugs-17-00218-t001:** NMR spectral data for **1–3** in acetone-*d*_6_.

Position	1					2					3				
	^1^H*^a^* (*J*,Hz)			^13^C*^b^*		^1^H*^a^* (*J*,Hz)			^13^C*^b^*		^1^H*^a^* (*J*,Hz)			^13^C*^b^*	
1a	1.55	dq	13.2, 3.6	25.9	(t)	1.52	dq	13.2, 3.6	25.9	(t)	1.80	dq	12.6, 3.6	26.0	(t)
1b	1.12	qd	13.2, 3.6			1.13	qd	13.2, 3.6			1.20	qd	12.6, 3.6		
2a	0.91	m		36.0	(t)	0.90	m		36.0	(t)	0.87	qd	12.6, 3.6	36.2	(t)
2b	1.74	ddt	12.6, 5.4, 3.6			1.75	ddt	12.6, 5.4, 3.6			1.74	ddt	12.6, 5.4, 3.6		
3	1.48	m		33.5	(d)	1.49	m		33.5	(d)	1.49	m		33.7	(d)
4a	0.82	q	12.6	42.6	(t)	0.83	q	12	42.7	(t)	0.79	q	12.6	43.0	(t)
4b	1.87	ddd	12.6, 5.8, 3.5			1.88	ddd	12.0, 5.4, 3.5			1.87	ddd	12.6, 5.4, 3.6		
5	1.92	br t	12.6	37.4	(d)	1.93	m		37.5	(d)	1.95	br t	12.6	37.5	(d)
6	5.55	d	10.2	133.2	(d)	5.54	d	10.2	132.9	(d)	5.57	d	9.6	133.1	(d)
7	5.70	ddd	10.2, 5.0, 2.2	125.3	(d)	5.69	ddd	10.2, 4.8, 2.4	125.7	(d)	5.97	ddd	9.6, 4.8, 3.0	126.1	(d)
8	2.65	ddt	11.8, 5.0, 1.8	49.9	(d)	2.74	ddt	12.0, 4.8, 1.8	49.7	(d)	3.34	ddt	12.6, 4.8, 0.9	45.3	(d)
9				52.9	(s)				52.9	(s)				53.4	(s)
10	1.49	m		38.7	(d)	1.40	td	13.2, 3.0	39.0	(d)	1.35	td	12.6, 3.6	40.4	(d)
11				211.6	(s)				211.7	(s)				211.3	(s)
12				74.0	(s)				74.2	(s)				74.7	(s)
13	3.16	dd	11.8, 9.0	52.8	(d)	3.01	dd	12.0, 9.0	52.5	(d)	2.83	ddd	12.6, 5.4, 2.4	45.5	(d)
14	5.62	ddd	15.5, 9.0, 1.8	124.3	(d)	5.82	ddd	15.0, 9.0, 1.8	125.5	(d)	6.42	ddq	8.8, 5.4, 2.4	133.6	(d)
15	5.73	dd	15.5, 5.5	141.9	(d)	5.59	ddd	15.0, 5.4, 0.6	140.8	(d)	5.83	ddq	8.8, 5.4, 2.4	134.3	(d)
16	4.21	m		67.8	(d)	4.17	m		68.0	(d)	2.48	m		43.7	(d)
17	1.16	d	6.6	24.2	(q)	1.14	d	6.6	23.9	(q)	1.20	d	7.2	15.3	(q)
18				207.1	(s)				207.8	(s)				81.2	(s)
19	3.66	d	6.8	70.4	(d)	3.93	d	4.2	69.7	(d)				173.3	(d)
20	3.99	m		67.9	(d)	3.98	m		67.4	(d)					
21	1.25	d	6.0	20.4	(q)	1.27	d	6.0	20.7	(q)					
22				170.7	(s)				170.2	(s)				177.5	(s)
23	0.91	d	6.6	22.6	(q)	0.92	d	6.6	22.6	(q)	0.89	d	6.6	22.8	(q)
24	0.99	s		16.0	(q)	0.99	s		16.1	(q)	1.05	s		17.3	(q)
16-OH	3.79	d	5.0			3.47	br s								
20-OH	4.17	d	6.0			4.02	br s								
NH	8.01	br s				8.02	br s								

**Table 2 marinedrugs-17-00218-t002:** NMR spectral data for **4** and **5** in acetone-*d*_6_.

Position	4					5				
	^1^H*^a^* (*J*,Hz)			^13^C*^b^*		^1^H*^a^* (*J*,Hz)			^13^C*^b^*	
1α	1.67	m		27.6	(t)	1.91	dq	12.6, 3.6	28.0	(t)
1β	1.20	qd	12.6, 3.6			1.20	qd	12.6, 3.6		
2α	0.85	qd	12.6, 3.6	36.1	(t)	0.90	m		36.2	(t)
2β	1.72	m				1.73	ddt	11.4, 5.4, 3.6		
3	1.49	m		33.5	(d)	1.49	m		33.5	(d)
4α	0.79	q	12.6	42.6	(t)	0.80	q	12	42.9	(t)
4β	1.85	ddd	12.6, 5.4, 3.6			1.87	ddd	12.0, 5.8, 3.5		
5	2.08	m		39.1	(d)	2.09	m		38.6	(d)
6	5.48	d	10.2	133.3	(d)	5.49	d	10.2	133.5	(d)
7	6.04	dd	10.2, 3.0	127.7	(d)	6.07	dd	10.2, 3.0	127.4	(d)
8				141.5	(s)				140.1	(s)
9				50.7	(s)				50.6	(s)
10	1.70	m		44.7	(d)	1.37	td	12.6, 3.6	45.3	(d)
11				208.4	(s)				205.2	(s)
12				70.5	(s)				72.7	(s)
13	3.71	d	9.6	43.6	(d)	3.96	d	9	41.5	(d)
14	5.52	d	1.8	124.6	(d)	5.70	d	1.8	126.9	(d)
15	5.37	ddq	16.8, 9.6, 1.8	128.4	(d)	5.37	ddq	16.8, 9.0, 1.8	129.4	(d)
16	5.57	dq	16.8, 6.6	131.0	(d)	5.62	dq	16.8, 6.6	129.9	(d)
17	1.62	d	6.6	18.0	(q)	1.59	d	6.6	17.9	(q)
18				208.4	(s)				204.7	(s)
19	3.49	d	7.2	69.9	(d)	3.81	d	4.8	68.9	(d)
20	3.99	m		68.1	(d)	3.90	m		67.0	(d)
21	1.28	d	6.6	20.2	(q)	1.29	d	6.0	20.8	(q)
22				168.1	(s)				170.7	(s)
23	0.89	d	6.6	22.6	(q)	0.89	d	6.6	22.6	(q)
24	1.12	s		17.4	(q)	1.12	s		16.6	(q)
20-OH	4.06	br s				4.02	d	6.0		
NH	7.88	br s				7.86	br s			

**Table 3 marinedrugs-17-00218-t003:** Cytotoxicity assay against P388 and HL-60 and L1210 cell lines.

Compounds	Cell line P388	Cell line HL-60	Cell line L1210
	IC_50_ (μM) *^a^*	IC_50_ (μM) *^a^*	IC_50_ (μM) *^a^*
**1**	36.2	21.5	22.1
**2**	20.0	12.1	8.0
**3**	61.2	41.6	27.1
**4**	9.7	6.1	8.4
**5**	15.5	6.2	10.3
5-Fluorouracil *^b^*	7.2	4.5	1.1

*^a^* DMSO was used as vehicle. *^b^* Positive control.

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
