# Peer review of "Altercrasins A–E, Decalin Derivatives, from a Sea-Urchin-Derived Alternaria sp.: Isolation and Structural Analysis Including Stereochemistry"

_marinedrugs, 2019, doi:10.3390/md17040218_

Reviewer 1 Report

The manuscript “Altercrasins A-E, decalin derivatives, from a sea-urchin-derived Alternia sp.: Isolation and structural analysis including stereochemistry” describes the isolation from a strain of Alternaria sp. OUPS-117D-1 of five altercrasins and some structural characterization of the four previously unreported compounds. Cytotoxic activities of the elucidated compounds have also been tested. The manuscript is properly organized and easy to read. Although the results sound feasible and might be useful, in my opinion the manuscript need some revision before publication in Marine Drugs.

Positive Comments:

 1.- The structural elucidation of altercrasins B and Cis robust and conclusive. The elucidation of altercrasins D and E seems the more than plausible based on the results presented.

2.- Presented data is abundant and supports the presented structures.

 Comments:

1.- Abstract: I consider that the abstract is not informative at all. According to the instructions for authors of the journal “The abstract should be a single paragraph and should follow the style of structured abstracts, but without headings: 1) Background: Place the question addressed in a broad context and highlight the purpose of the study; 2) Methods: Describe briefly the main methods or treatments applied. Include any relevant preregistration numbers, and species and strains of any animals used. 3) Results: Summarize the article's main findings; and 4) Conclusion: Indicate the main conclusions or interpretations.”. I miss most of the required information for an abstract. Please, rewrite.

 2.- Some sentences requires appropriate references:

a) Lines 22-23: “Marine organisms are potential…. For the development of new chemotherapy agents”

b) Lines 32-33: “The absolute configuration … unless a good single crystal is obtained”

c) Lines 33-35: “However, in a previously reported study… by experiments via a chemical transformation”

3.- Since the structure of altercrasin B is mostly discussed by comparison with altercrasin A, I consider that the explanation of the elucidation of altercrasins B will be clearer if Figure 2 also contains the results for the 2D NMR correlations in altercrasin A. Please, consider to add.

4.- Line 108. The sentence “in addition to some contradiction of NOESY correlation” requires a more detailed explanation. Please, add

5.- Although the structural elucidation of altercrasins D and E is less conclusive than the previous compounds, I consider that the proposed structures are the most plausible based on the results presented. However, I would like to see a Figure showing the 2D NMR correlations (as Figure 2) but for these two compounds.

6.- In my opinion, the section related with cytotoxicity could be improived by including the positive control results in the discussion. The authors only discuss the comparison between the different compounds.

 7.- MS results: All compounds contain a nitrogen atom, it is estrange that the authors used the [M+Na]+ for the molecular formula establishment of the four isolated compounds but [M+H]+ for the one previously reported. Please, comment.

8.- The section Assay for cytotoxicity is missed. I could only see a paragraph likely coming from the non-edited version of the layout. Please, include

Author Response

marinedrugs-480817

Revised detail

Reviewer 1

1.- Abstract: I consider that the abstract is not informative at all. According to the instructions for authors of the journal “The abstract should be a single paragraph and should follow the style of structured abstracts, but without headings: 1) Background: Place the question addressed in a broad context and highlight the purpose of the study; 2) Methods: Describe briefly the main methods or treatments applied. Include any relevant preregistration numbers, and species and strains of any animals used. 3) Results: Summarize the article's main findings; and 4) Conclusion: Indicate the main conclusions or interpretations.”. I miss most of the required information for an abstract. Please, rewrite.

Response: As be pointed out, we rewrote the abstract according to the instructions. For background, method, result, and conclusion, we added the required information.

2.- Some sentences require appropriate references:

a) Lines 22-23: “Marine organisms are potential…. For the development of new chemotherapy agents”

b) Lines 32-33: “The absolute configuration … unless a good single crystal is obtained”

c) Lines 33-35: “However, in a previously reported study… by experiments via a chemical transformation”

Response: We added newly ref. 1 and 2, and inserted references where pointed out, respectively.

3.- Since the structure of altercrasin B is mostly discussed by comparison with altercrasin A, I consider that the explanation of the elucidation of altercrasins B will be clearer if Figure 2 also contains the results for the 2D NMR correlations in altercrasin A. Please, consider to add. 

Response: Thank you for your suggestion. There is almost no difference between the 1 and 2 of 2D NMR correlations; therefore, we think that the insertion to Figure 2 of 2D data of 1 is not very useful.

4.- Line 108. The sentence “in addition to some contradiction of NOESY correlation” requires a more detailed explanation. Please, add

Response: The following sentences were added as a supplement.             

i.e., the correlation between H-17 and H-13 should not be observed in 3a.”

5.- Although the structural elucidation of altercrasins D and E is less conclusive than the previous compounds, I consider that the proposed structures are the most plausible based on the results presented. However, I would like to see a Figure showing the 2D NMR correlations (as Figure 2) but for these two compounds.

Response: As be pointed out, we added the 2D NMR correlations of altercracin C (3) and D (4) as Figure 4 and 7, respectively. In the 4 and 5 of 2D NMR correlations, there is almost no difference as those of 1 and 2.

6.- In my opinion, the section related with cytotoxicity could be improved by including the positive control results in the discussion. The authors only discuss the comparison between the different compounds. 

Response: As be pointed out, we added the following sentences as the discussion including the comparison with the positive control results.

In especially, the activity of 4 was equal to that of 5-fluorouracil.”

7.- MS results: All compounds contain a nitrogen atom, it is estrange that the authors used the [M+Na]+ for the molecular formula establishment of the four isolated compounds but [M+H]+ for the one previously reported. Please, comment.

Response: Thank you for your opinion. I mistook some of their MS data. Those of 2, 4, and 5 were replaced [M+Na]+ data to [M+H]+ data; however, the high resolution MS of 3 was only measured [M+Na]+. There is no particular reason.

8.- The section Assay for cytotoxicity is missed. I could only see a paragraph likely coming from the non-edited version of the layout. Please, include

Response: I am sorry that the methods for assay for cytotoxicity was missing. We inserted it to the section entitled 3.4.

Reviewer 2 Report

The review concerns the Manuscript ID: marinedrugs-480817. Title: Altercrasins A–E, decalin derivatives, from a sea-urchin-derived.

The manuscript can be accepted for publication, however, the following correction should be done:

1. Figure 1. The structure are too small, in particular, skeleton notation numbers. Please, make them a little more bigger. There is place for that.

2. Supporting Information. Each 1H and 13C NMR spectrum should be on separate page in vertical position to maximize the area of spectrum, as in case of FABS spectra. In present form the spectra are too small, and thus, it is difficult to read them. Positions of the signals should be also depicted.

3. Integration is incorrectly fixed. For example, Figure S1, the integration of single protons 8 and 13 (2.65 and 3.16 ppm) are integrated as 2.41-2.42, amide proton (8.01 ppm) integrated as 1.95. For olephinic sp2 protons (5.55-5.73 ppm) integrated as 11.16. Similar remark is for the spectra of remaining compounds.

4. DEPT spectrum, mentioned in the main text should be also introduced in Supporting Information.

Minor remark. There is an atypical sign in the title in the fragment “a sea”.

Author Response

marinedrugs-480817

Revised detail

Reviewer 2

The review concerns the Manuscript ID: marinedrugs-480817. Title: Altercrasins A–E, decalin derivatives, from a sea-urchin-derived.

The manuscript can be accepted for publication, however, the following correction should be done:

Figure 1. The structures are too small, in particular, skeleton notation numbers. Please, make them a little bigger. There is place for that.

Response: We enlarged Figure 1 as a whole.

Supporting Information. Each 1H and 13C NMR spectrum should be on separate page in vertical position to maximize the area of spectrum, as in case of FABS spectra. In present form the spectra are too small, and thus, it is difficult to read them. Positions of the signals should be also depicted.

Response: As be pointed out, we replaced each 1H and 13C NMR spectrum into separate page in vertical position. In addition, we inserted some zoomed images of the NMR spectra to Supporting Information.

Integration is incorrectly fixed. For example, Figure S1, the integration of single protons 8 and 13 (2.65 and 3.16 ppm) are integrated as 2.41-2.42, amide proton (8.01 ppm) integrated as 1.95. For olephinic sp2 protons (5.55-5.73 ppm) integrated as 11.16. Similar remark is for the spectra of remaining compounds.

Response: Thank you for your pointing out. Traces of mixtures and straight integral curves often affect the integrated values. Under sufficient purity conditions, the integrated value in 1H NMR chart were used as a guide only.

DEPT spectrum, mentioned in the main text should be also introduced in Supporting Information.

Response:  As be pointed out, we inserted each DEPT to Supporting Information.

Minor remark. There is an atypical sign in the title in the fragment “a sea”.

Response: Thank you for your pointing out. We removed an extra space in front of “a sea”.

Reviewer 3 Report

In the present manuscript Yamada et al. report on the structural and stereochemical characterization of four metabolites isolated from the fungus Alternaria sp.  The data are exhaustive and the studies done for the assignment of the absolute configuration seem thorough and accurate. In addition to the structural analysis, an assay to establish the cytotoxicity of the extracted compounds is reported. This analysis, although limited, interestingly highlight that the difference in stereochemistry has no impact on the activity of the compounds.

In my opinion this article can be of interest for the scientific community working on natural products and interested in the development of new cytotoxic scaffold, therefore I recommend it for publication in Marine drugs after the following minor revisions are provided by the authors:

1. Line 28 please change “as has been reported…” with “as it has been reported” and put “1” in bold

2.Line 48-49: please substitute “as a pale yellow oil” with “as pale yellow oils”

3. Line 50: please change “as has been observed…” with “as it has been observed”

4. The sentence starting at line 95 and finishing at line 97 is not very clear, I would suggest to rewrite it. The same for the sentence at line 133-134.

5. Lines 200-203 seem to me to have no correlation with the section entitled 3.4, while the methods for assay for cytotoxicity are missing.

6. Please, write in the supporting information the data relative to the instruments used.

7. Please insert in the supplementary also zoomed images of the NMR spectra, this would be of great help in retrieving the correlations explained in the manuscript

Author Response

marinedrugs-480817

Revised detail

Reviewer 3

In the present manuscript Yamada et al. report on the structural and stereochemical characterization of four metabolites isolated from the fungus Alternaria sp.  The data are exhaustive and the studies done for the assignment of the absolute configuration seem thorough and accurate. In addition to the structural analysis, an assay to establish the cytotoxicity of the extracted compounds is reported. This analysis, although limited, interestingly highlight that the difference in stereochemistry has no impact on the activity of the compounds.

In my opinion this article can be of interest for the scientific community working on natural products and interested in the development of new cytotoxic scaffold, therefore I recommend it for publication in Marine drugs after the following minor revisions are provided by the authors:

Line 28 please change “as has been reported…” with “as it has been reported” and put “1” in bold

Response: Thank you for your pointing out, we put “1” in bold.

    2.  Line 48-49: please substitute “as a pale yellow oil” with “as pale yellow oils”

Response: Thank you for your pointing out, we corrected to “as pale yellow oils”.

     3.  Line 50: please change “as has been observed…” with “as it has been observed”

Response: Thank you for your pointing out, we changed to “as it has been observed”.

    4. The sentence starting at line 95 and finishing at line 97 is not very clear, I would suggest to rewrite it. The same for the sentence at line 133-134.

Response: At line 95-97, Figure 3 (2D NMR correlations of 3) was added, and the sentence was rewrote as shown in line 104-106.

Response: At line 133-134, the sentence in not very clear as be pointed out. We changed the order of the explanation as shown in line 147-150 to make it clear.

5. Lines 200-203 seem to me to have no correlation with the section entitled 3.4, while the methods for assay for cytotoxicity are missing.

Response: I am sorry that the methods for assay for cytotoxicity was missing. We inserted it to the section entitled 3.4.

   6.   Please, write in the supporting information the data relative to the instruments used.

Response: the data relative to the instruments used is described section 3.1. General Experimental Procedures.

7. Please insert in the supplementary also zoomed images of the NMR spectra, this would be of great help in retrieving the correlations explained in the manuscript

Response: As be pointed out, we inserted some zoomed images of the NMR spectra including each DEPT to Supporting Information.

Round  2

Reviewer 1 Report

The revised version of manuscript entitled “Altercrasins A-E, decalin derivatives, from a sea-urchin-derived Alternia sp.: Isolation and structural analysis including stereochemistry” addressed most of my comments. In my opinion the current version of the manuscript deserves publication in Marine Drugs.

My only comment is to write the number of the correct derivative in the caption of the new Figure 4.

Author Response

To Reviewer 1

Thank you for careful review.

As be pointed out, we corrected the number of the derivative in the caption of the new Figure 4.